# Progress and Future Directions with Peptide-Drug Conjugates for Targeted Cancer Therapy

**DOI:** 10.3390/molecules26196042

**Published:** 2021-10-05

**Authors:** Jakob Lindberg, Johan Nilvebrant, Per-Åke Nygren, Fredrik Lehmann

**Affiliations:** 1Oncopeptides AB, Västra Trädgårdsgatan 15, SE-111 53 Stockholm, Sweden; jakob.lindberg@oncopeptides.com; 2Department of Protein Science, School of Engineering Sciences in Chemistry, Biotechnology and Health, KTH-Royal Institute of Technology, SE-100 44 Stockholm, Sweden; johanni@kth.se (J.N.); perake.nygren@biotech.kth.se (P.-Å.N.); 3SciLifeLab, SE-171 65 Solna, Sweden

**Keywords:** peptide-drug conjugates, cancer, tumor-targeting drug conjugates, cytotoxic agent

## Abstract

We review drug conjugates combining a tumor-selective moiety with a cytotoxic agent as cancer treatments. Currently, antibody-drug conjugates (ADCs) are the most common drug conjugates used clinically as cancer treatments. While providing both efficacy and favorable tolerability, ADCs have limitations due to their size and complexity. Peptides as tumor-targeting carriers in peptide-drug conjugates (PDCs) offer a number of benefits. Melphalan flufenamide (melflufen) is a highly lipophilic PDC that takes a novel approach by utilizing increased aminopeptidase activity to selectively increase the release and concentration of cytotoxic alkylating agents inside tumor cells. The only other PDC approved currently for clinical use is ^177^Lu-dotatate, a targeted form of radiotherapy combining a somatostatin analog with a radionuclide. It is approved as a treatment for gastroenteropancreatic neuroendocrine tumors. Results with other PDCs combining synthetic analogs of natural peptide ligands with cytotoxic agents have been mixed. The field of drug conjugates as drug delivery systems for the treatment of cancer continues to advance with the application of new technologies. Melflufen provides a paradigm for rational PDC design, with a targeted mechanism of action and the potential for deepening responses to treatment, maintaining remissions, and eradicating therapy-resistant stem cells.

## 1. Introduction

The development of targeted therapies that selectively bind and act through molecules expressed by cancer cells has been a major advancement in the treatment of cancer, as they mediate higher efficacy with enhanced tolerability compared with traditional cytotoxic agents [1]. Conjugated drugs, combining a tumor-selective moiety with a cytotoxic agent, are effective in delivering the cytotoxic payload directly to the tumor cell, thereby ameliorating the systemic side effects usually associated with cytotoxic chemotherapy [1,2]. 

In cancer therapy, there are three targeted approaches used to enhance the specificity and antitumor activity of a cancer treatment (Figure 1). The targeted agent can be designed to inhibit a protein, such as a receptor or enzyme, expressed by the tumor cell and required for tumor cell growth and progression. Another approach is for an effector molecule, such as an antibody-drug conjugate (ADC), bispecific antibody, or chimeric antigen receptor T-cell therapy, to bind to a molecule that is overexpressed on the tumor cell surface, and provide synergistic inhibition of tumor cell division along with the delivery of a cytotoxic payload or stimulation of a tumor-directed immune response [1]. The third approach is to use a peptide-drug conjugate (PDC) with a cleavable linker that has a peptide moiety targeting the tumor cell and drives enrichment of toxic payload in tumor cells. Aminopeptidases represent one rational target in tumor cells for PDCs, catalyzing the hydrolysis of terminal amino acid residues from proteins or peptides and operating downstream of the ubiquitin–proteasome system [1,2]. PDCs targeting peptide hormone receptors that are overexpressed on tumor cells are another mechanism for delivering a cytotoxic payload into the tumor [3].

Many of the targeted agents currently used clinically are based on monoclonal antibodies; however, more recently, the benefits of peptides as tumor-homing agents have been explored. In this review, we discuss drug conjugates as cancer treatments, touching briefly on the strengths and limitations of a group of clinically approved ADCs, before focusing on recent advances with PDCs. In particular, we discuss the two PDCs that have been approved by the FDA for clinical use as cancer treatments: melphalan flufenamide (melflufen) and ^177^Lu-dotatate. Melflufen is a PDC that targets aminopeptidases and thereby rapidly releases alkylating agents inside tumor cells [4,5,6,7,8]. ^177^Lu-dotatate has a different mechanism of action. ^177^Lu-dotatate is a peptide receptor radionuclide therapy that is a targeted form of radiotherapy, allowing the delivery of a radionuclide payload directly to tumor cells that are expressing receptors and binding the homing peptide [3]. We also briefly cover some selected examples of PDCs currently under clinical development. 

## 2. Structure of Anticancer Drug Conjugates

A range of approaches is shown to be effective in the selective delivery of different effector molecules to cancer cells. Conjugated drugs developed for targeted drug delivery usually consist of three components, all of which contribute to the overall biological efficacy and selectivity of the drug (Figure 2 [1]). A carrier moiety is included in the structure, which targets a tumor-specific marker. Besides peptides, a range of other small molecules and biologics such as native proteins, antibodies, affibodies, designed ankyrin repeat proteins, and aptamers have been investigated for providing tumor selectivity [9,10,11,12,13].

The carrier molecule is linked to an active anticancer agent, which can induce a variety of biological functions. Currently, drug conjugates developed for cancer treatment include predominantly cytotoxic molecules and radionuclides for mediating cytotoxicity. The covalent linker that connects the targeting carrier molecule with the effector molecule may be cleavable or non-cleavable. Cleavable linkers enable controlled drug release of the payload once the conjugate has bound to or entered the tumor cell. The choice of cleavable or non-cleavable linkers depends on the requirements for the design and mechanism of action of the targeted therapy [2].

## 3. Limitations with Antibody-Drug Conjugates as Cancer Treatments

Currently, in oncology, ADCs are the most common drug conjugates used clinically as cancer treatments [2]. A broad range are approved for use in a variety of hematologic malignancies and, less frequently, for solid tumors (Table 1). The strength of ADCs is that they incorporate a tumor-homing monoclonal antibody with high affinity for the specific antigen differentially expressed by cancer cells. For example, a few ADCs incorporate trastuzumab, a humanized monoclonal antibody targeting the human epidermal growth factor receptor-2 (HER2). Trastuzumab in itself has antitumor activity and has been used as treatment for HER2-positive breast cancer for more than 20 years [14]. Conjugates combining trastuzumab with small cytotoxic molecules enhance and broaden antitumor activity. For example, trastuzumab emtansine is comprised of trastuzumab conjugated via a non-cleavable maleimidomethyl cyclohexane-1-carboxylate thioether linker to the cytotoxin DM1 [15]. DM1 induces cell death by inhibiting tubulin polymerization. DM1 alone has a narrow therapeutic window, but its linkage to trastuzumab, with an average of 3.5 linker-payload molecules per antibody, selectively targets the cytotoxin to HER2-positive cancer cells, thereby widening its therapeutic window [16]. The antitumor activity of trastuzumab emtansine is mediated in a number of ways, including the HER2 signaling blockade, Fc-mediated immune response, and DM1-mediated inhibition of cell division [16]. The broader activity manifests clinically with trastuzumab emtansine being active in patients with metastatic breast cancer who have received multiple lines of HER2-directed therapy [17].

The therapeutic application of ADCs is limited by their physiochemical and pharmacodynamic properties [1,2]. Monoclonal antibodies are large macromolecules. The high molecular weight of ADCs (around 150 kDa; Table 1 [18,19,20,21,22,23,24,25,26,27,28,29,30,31,32]), along with their protein-like physiochemical properties, limits the distribution of ADCs into tumor tissue [1,33]. In particular, ADCs have limited utility/efficacy in solid tumors, where they fail to penetrate through the vasculature and into the tumor tissue. Another limitation of ADCs is that monoclonal antibodies, even when humanized, frequently are immunogenic and lead to the formation of anti-drug antibodies that can interfere with their pharmacokinetic and pharmacodynamic properties. The prescribing information for the ADCs listed in Table 1 indicates that anti-drug antibodies have been detected in patients receiving ADCs (ranges varying from 10% to up to 70% [18,19,20,21,22,23,24,25,26,27,28,29,30,31,32]). The clinical consequences of the formation of anti-drug antibodies are often unclear; however, studies with brentuximab vedotin showed that patients who developed persistently positive anti-drug antibodies experienced a higher incidence of infusion-related reactions [24]. In addition to anti-drug antibodies, the expression pattern of the target antigen for a homing antibody in an ADC can lead to ‘on-target, off-tumor’ toxicities that may or may not be related to the cytotoxicity of the payload [33], although this can be true of any tumor-homing carrier. On a practical level, the complex structure of ADCs requires cell-based production systems, which are associated with high production costs. These higher costs can limit patients’ access to targeted treatments, due to payers’ and healthcare providers’ concerns about the cost-effectiveness of these treatments [1,2]. Another challenge is that many cytotoxic drugs are hydrophobic and tend to induce antibody aggregation, which can lead to faster clearance rates and immunogenicity [34,35].

## 4. Peptide-Drug Conjugates as Targeted Cancer Treatments

Some of the limitations of ADCs can be overcome by using smaller biomolecules, such as peptides, in drug conjugates. PDCs have a similar structure to ADCs but use a peptide moiety to preferentially target the drug conjugate to the tumor cell and limit off-target cytotoxicity by releasing the cytotoxic payload at the tumor site or within the tumor, rather than in healthy tissues. The definition of a peptide, as proposed by the US Food and Drug Administration (FDA), is a polymer composed of 40 or fewer amino acids [2,36]. There are numerous benefits associated with the use of peptides as carrier molecules in drug conjugates [1,2,6]; they are easy to synthesize and provide well-defined and cost-effective targeted treatments compared with ADCs. Structural modifications can be easily introduced, supporting rational drug design to increase bioavailability, binding affinity, and stability. The incorporation of lipids/fatty acids enhances the lipophilicity of the peptide, which can tailor the half-life and bioavailability by modulating tumor penetration and cellular uptake [1]. Peptides are also amenable to combinatorial drug discovery and support high-throughput screening in vitro of candidate structures in order to identify peptides with optimized pharmacodynamic properties. In addition, peptides have low intrinsic immunogenicity and, similar to antibodies and other protein macromolecules, are metabolized to non-toxic components [1,37].

Within a PDC, the properties of the peptide carrier can be adapted to fine-tune the mechanism of action to be relevant to the target. The peptides used in PDCs are divided into two categories: cell-penetrating peptides (CPPs) and cell-targeting peptides (CTPs) [1]. Some PDCs that include a cell-penetrating homing peptide enter the cell via a non-specific mechanism (i.e., CPPs), whereas others include a peptide that specifically binds to an antigen or receptor on the surface of the tumor cell to mediate delivery of the cytotoxic payload in the vicinity of or into the tumor cell (i.e., CTPs) [1]. While increased drug delivery is associated with proteins attached to CPPs, the extensive use of these types of PDCs has been limited due to their low cell specificity. In contrast, CTPs bind with a high affinity to overexpressed receptors on the surface of the tumor cell, displaying a similar action as monoclonal antibodies while overcoming certain disadvantages observed with monoclonal antibodies (e.g., limited tissue penetration and requirement for cell-based production systems) [1]. Melflufen has a novel mechanism of action that combines a cell-penetrating peptide with a tumor-targeting mechanism of action (Figure 3) [38]. Melflufen (melphalan flufenamide; C_24_H_30_C_l2_FN_3_O_3_; l-melphalanyl-p-l-fluorophenylalanine ethyl ester hydrochloride; previously denoted J1) is an ethyl ester of a dipeptide combining melphalan with *para*-fluoro-L-phenylalanine. It was designed to exploit the overexpression of peptidases and esterases by cancer cells. Melflufen is a highly lipophilic PDC (Figure 3a) that rapidly and passively crosses the cell membrane [6]. Once inside the tumor cell, the aminopeptidase-binding domain of melflufen is a substrate for aminopeptidases and esterases and releases entrapped hydrophilic alkylator payloads [4,6,38] (Figure 3b). Alkylating agents prevent cell division and cause cell death via the modification of DNA, triggering DNA fragmentation, intra- and inter-strand DNA cross-linking, or DNA mutations due to nucleotide mispairing [39,40]. 

The properties of melflufen have two important implications for cancer treatment. Firstly, melflufen is a substrate for aminopeptidases and esterases that are overexpressed by tumor cells. This feature results in the preferential accumulation of melflufen and its alkylating payloads in tumor cells [6]. Secondly, the high load of alkylating moieties in tumor cells triggers rapid and irreversible DNA damage. It has been proposed that the accumulation of a high concentration of alkylating agents within tumor cells following administration of melflufen supports its ability to overcome melphalan resistance in both clinical and nonclinical studies [5,8]. 

The enzyme-catalyzed hydrolysis of melflufen is an efficient and quick process, with the specificity of the intracellular delivery confirmed by the inhibition by the aminopeptidase inhibitor bestatin and the esterase inhibitor ebelactone [6]. Aminopeptidases are Zn^2+^-dependent metalloproteinases that catalyze the hydrolysis of amino acids at the N-terminal position from oligopeptides and have been associated with multiple tumorigenic processes such as proliferation, apoptosis, differentiation, angiogenesis, and motility [41,42]. Aminopeptidases have been identified as therapeutic cancer targets, with attempts to treat cancer by direct inhibition of aminopeptidase activity reported [43,44]. Melflufen takes a novel approach by utilizing increased aminopeptidase expression to selectively release potent alkylating agents inside tumor cells. A range of aminopeptidases expressed in multiple myeloma and other tumors can mediate the hydrolytic cleavage and rapid release of the cytotoxic alkylator payloads from melflufen, including aminopeptidase N, LAP3, LTA4H, RNPEP, and aminopeptidase B [38,41].

^177^Lu-dotatate (C_65_H_90_N_14_O_19_S_2_; (177)Lu-[DOTA(0),Tyr(3)] octreotate; Figure 4) is a peptide receptor radionuclide therapy that is a targeted form of radiotherapy combining a somatostatin analog with a radionuclide, linked via the chelating agent DOTA (which consists of a central 12-membered tetraaza ring) [3]. The somatostatin receptor is expressed in over 80% of well-differentiated neuroendocrine tumors, which offers a rational target for a PDC including a somatostatin analog as a homing peptide. Lutetium-177 is a beta- and gamma-emitting radionuclide with a maximum particle range of 2 mm and a half-life of ~160 hours [3]. ^131^I-tositumomab is an intact anti-CD20 antibody conjugated to a radionuclide, which is approved by the FDA for the treatment of non-Hodgkin lymphoma [45]. These examples demonstrate the therapeutic benefit of radionuclide therapy in the treatment of a variety of malignancies. 

## 5. The Clinical Application of Peptide-Drug Conjugates in Cancer

As discussed above, much of the research and development evaluating drug conjugates as anti-cancer therapies has focused on ADCs. The main barrier for the development of PDCs was the rapid clearance of peptides that limited their therapeutic utility [2]; however, advances have led to the development of chemical and structural modifications of peptides that have significantly improved their bioavailability [2,6,46]. There are many PDCs in clinical development with features designed to enhance enzymatic and chemical stability that have been the subject of recent reviews [2,46]. Peptide modifications that increase stability and improve pharmacokinetics of peptides include replacing L-amino acids with D-amino acids, modification of amino acid side chains, and peptide cyclization, in order to reduce susceptibility for proteases and reduce metabolism. Renal clearance of peptides can be slowed down by the modification of N- and C-termini, conjugation with other molecules to increase lipophilicity and albumin binding, and the addition of fatty acid chains or PEGylation. The formulation of peptides with permeation enhancers and coatings can enhance oral bioavailability [2]. Despite their potential and the high level of research and development, currently, only two PDCs have been approved by the FDA for clinical use as cancer treatments: melflufen and ^177^Lu-dotatate (Table 1). 

Melflufen is approved, in combination with dexamethasone, for the treatment of heavily pre-treated adult patients with relapsed or refractory multiple myeloma (RRMM [18]). The accelerated approval of melflufen plus dexamethasone by the US FDA was based on the results of the Phase II HORIZON study in patients with heavily pre-treated, resistant, and high-risk RRMM [47]. Melflufen achieved an overall response rate (ORR) of 29% in the overall population, with an ORR of 26% in patients with triple-class refractory MM. Melflufen was active in difficult-to-treat patients with MM resistant to multiple drug classes, high-risk cytogenetics, and/or extramedullary disease. Triplet combinations with melflufen plus dexamethasone combined with either daratumumab or bortezomib are promising in patients with RRMM [48]. Top-line results from the Phase III OCEAN study (NCT03151811) showed that melflufen plus dexamethasone was superior for progression-free survival (PFS) versus the standard-of-care of pomalidomide plus dexamethasone in patients with RRMM [49].

Melflufen also shows promise as a cancer treatment for tumor types beyond MM. Preclinical studies show that melflufen has enhanced cytotoxic activity versus melphalan in a wide range of human cancer cell lines, including hematologic malignancies, neuroblastoma, lung cancer (small and non-small cell), ovarian, renal, and breast cancer [6]. A recent study investigated the in vitro and in vivo efficacy of melflufen on isogenic normal and HER2-transformed tumorigenic breast epithelial lines [50]; the tumorigenic breast cancer cell line was almost 10-fold more sensitive to melflufen than the normal breast epithelial line. The antitumor activity of melflufen was attenuated by the addition of the aminopeptidase inhibitor bestatin [50]. Preclinical studies have shown the potential for melflufen to affect multiple cellular processes involved in cancer growth and progression, including metastatic processes and angiogenesis, as well as demonstrating efficacy in models of multidrug resistance, and in the presence of *TP53* deletion [51,52]. 

The approval of ^177^Lu-dotatate was based on the results of the Phase III NETTER-1 study, which showed that ^177^Lu-dotatate given every 8 weeks (four doses overall) plus the best supportive care, including long-acting repeatable octreotide acetate (LAR), achieved statistically significant improvements in progression-free survival, ORR, and overall survival compared with LAR alone in patients with well-differentiated, metastatic midgut neuroendocrine tumors [53]. In addition to improving outcomes for patients with midgut neuroendocrine tumors, ^177^Lu-dotatate has established proof of concept for the clinical benefits that can be provided by peptide receptor radionuclide therapy. 

More biological targets have been identified that could support peptide-guided radionuclide therapy with ^177^Lu in other tumor types. One promising peptide target is the prostate-specific membrane antigen (PSMA) in prostate cancer [54]. PSMA is a transmembrane glycoprotein that is expressed in healthy prostate tissue, with increasing levels of expression seen in prostate cancer as it progresses to higher grades and metastatic progression. PSMA is a highly attractive target for both imaging and therapy in prostate cancer. ^177^Lu-PSMA-617 is a peptide receptor radionuclide therapy comprised of PSMA-617, a small molecule that binds with high affinity to PSMA via the Glu-urea PSMA binding motif, linked via a DOTA chelator to ^177^Lu-dotatate [55]. Upon binding to PSMA on the cell membrane, ^177^Lu-PSMA-617 is internalized and the emission of beta particles within the cell damages tumor DNA and leads to cell death [54]. Based on the results of early clinical studies, ^177^Lu-PSMA-617 has been used on a compassionate-use basis for patients with advanced metastatic castration-resistant prostate cancer who have exhausted all available treatment options [54]. A single-arm Phase II study showed that ^177^Lu-PSMA-617 achieved objective responses in nodal or visceral disease in 14 (82%) of 17 patients with measurable disease, with improvements in pain severity and global health scores [56]. More recently, the results of the VISION Phase III study evaluating ^177^Lu-PSMA-617 versus the best standard of care in patients with progressive PSMA-positive metastatic castration-resistant prostate cancer were reported [57]. The trial met both primary endpoints of overall survival and radiographic progression-free survival [57], and it is anticipated that the results of this study will be submitted to health authorities and support approval of ^177^Lu-PSMA-617 by the end of 2021. 

Currently in clinical development are two PDCs targeting Sortilin 1 (SORT1) receptors: TH1902 and TH1904 [58]. SORT1 receptors are overexpressed in several malignancies, including breast cancer and ovarian cancer. TH1902 is a PDC with a docetaxel payload, which received Fast Track designation from the FDA for the treatment of patients with sortilin positive recurrent advanced solid tumors that are refractory to standard therapy. TH1902 is currently being investigated in a Phase I clinical trial [58], and TH1904 contains a doxorubicin payload, and is currently under preclinical investigation [58].

Other PDCs in clinical development include synthetic analogs of natural peptide ligands linked with cytotoxic chemotherapy agents, including doxorubicin and paclitaxel, as reviewed recently by He et al. [46]. Synthetic analogs of natural peptide ligands are of particular interest for PDCs, as they can possess high target affinity and specificity, fast internalization rates, and low immunogenicity [1]. To date, results with these PDCs have been mixed, indicating that some challenges remain in translating the favorable pharmacodynamic properties of rationally designed PDCs seen in nonclinical and early clinical studies into improved clinical outcomes for patients (Table 2).

## 6. Future Directions with Peptide-Drug Conjugates 

The field of drug conjugates as drug delivery systems for the treatment of cancer continues to advance with the application of new technologies supporting the development of new PDC modalities. 

Affibodies (comprised of 58 amino acids) and albumin-binding domain-derived affinity proteins (ADAPTs; comprised of 46 amino acids) are two classes of polypeptides that fold into stable three-helix bundle structures and can be engineered for selective high-affinity binding to a variety of target structures, including cell surface receptors on tumor cells. Although slightly larger than the FDA definition of peptides [36], these units are sufficiently small to be efficiently produced via synthetic routes for subsequent site-specific conjugation to drugs and radionuclide chelators. This makes them highly promising for use as targeting units in future PDCs [12,63], particularly due to their typically high affinity, easy production, and characteristic control of the drug molecules’ loading and spatial arrangement [64].

Bicycle-toxin conjugates (BTCs) are another new modality. The tumor-homing moiety is a synthesized bicycle peptide (9–20 amino acids) including three cysteine residues that react with a small molecule linker to maintain the peptide in a rigid conformation, even when bound to a cytotoxic payload [2]. BTCs can deliver a toxin payload to the tumor cell. As with other PDCs, the advantages of BTCs over ADCs include enhanced tumor penetration, rapid extravasation, and slower renal clearance [2]. A number of BTCs are in early clinical development. BT5528 comprises a bicyclic peptide targeting the tumor antigen EphA2 linked to a cytotoxin (monomethyl auristatin E) via a tumor microenvironment cleavable linker [2]. A Phase I/II study of BT5528 in patients with recurrent advanced solid tumors expressing EphA2 (NCT04180371) started in Q4 2019 [65,66].

Peptide dendrimer conjugates are another promising drug delivery modality. Dendrimers are nanoparticles that can encapsulate a cytotoxic drug. They consist of globular molecules constructed from branched layers [67]. The modification of the dendrimer surface with functional groups or peptide sequences targeting receptors or other cell-surface antigens can be used to create ‘intelligent nanoparticles’ that target the cancer cell [2,67,68]. A peptide dendrimer conjugate has been designed for the active targeting and specific delivery of therapeutic agents into colorectal cancer cells. It comprises carboxymethylchitosan/poly(amidoamine) (CMCht/PAMAM) dendrimer nanoparticles functionalized with YIGSR laminin receptor-binding peptide. Preclinical studies confirmed that gemcitabine-loaded YIGSR-CMCht/PAMAM dendrimer nanoparticles are cytotoxic when tested against a colon cancer cell line, with no cytotoxicity observed when the cells were incubated with the dendrimer alone (without gemcitabine loading). Some cytotoxicity occurred when a fibroblast cell line lacking the laminin receptor was incubated with the gemcitabine-loaded YIGSR-CMCht/PAMAM dendrimer nanoparticles, indicating that some leakage of the cytotoxic agent from the dendrimer occurs in vitro [67,68]. While challenges remain, the research and development of nanotechnologies, particularly dendrimer nanoparticles, continues at a fast pace due to their promise to provide multifunctional and personalized cancer therapeutics [67,68].

Self-assembling PDCs are an emerging subset of PDCs, in which the conjugates are able to form nanostructures with unique physicochemical properties that differ from those of the individual components. Thus, self-assembling PDCs can form a drug delivery vehicle with the ability to breakdown, either over time or as a result of specific stimuli, in order to release the active pharmaceutical. Self-assembling PDCs can help to avoid premature degradation and rapid clearance of the active pharmaceutical [69]. For example, self-assembling camptothecin-based and paclitaxel-based PDCs have been associated with high drug loading and excellent stability [69,70,71]. Self-assembling PDCs can passively target and enhance the accumulation of the active pharmaceutical at tumor sites via the enhanced permeability and retention effect [72].

## 7. Conclusions and Perspectives on Progress with Drug Conjugates as Cancer Therapeutics

This review finds that the promise of drug conjugates to provide effective and tolerable targeted drug delivery systems in patients with cancer has been realized. ADCs targeting tumor antigens on the cell surface have provided robust proof of concept, with several ADCs approved at the current time predominantly for the treatment of hematologic cancers. PDCs have the potential to overcome some limitations of ADCs, but many have failed to realize their potential in clinical studies. This indicates that challenges remain in translating the rational design of drug conjugates, including tumor-homing peptides with cytotoxic agents, into effective therapies for patients with cancer. Despite this, there have been some successes, namely with melflufen and ^177^Lu-dotatate, both approved as cancer treatments by the FDA. 

Melflufen is a first-in-class PDC that targets cells with high aminopeptidase activity and rapidly releases alkylating agents inside tumor cells [4,5,6]. It was approved in the USA based on its clinical activity in heavily pretreated patients with RRMM. For these patients, this first-in-class PDC provides a potent targeted agent with a unique mechanism of action [6]. As a highly lipophilic entity that is rapidly taken up by tumor cells overexpressing aminopeptidases, the resulting preferential accumulation of melflufen and a high concentration of the alkylating payloads in tumor cells triggers irreversible DNA damage. It has been shown that this PDC agent offers potential for deepening responses to treatment, maintaining remissions, eradicating therapy-resistant stem cells, and, ultimately, improving outcomes for patients with MM [5,8,47]. Additionally, ^177^Lu-dotatate is a novel and effective targeted radiotherapy for patients with midgut neuroendocrine tumors, and has established proof of concept for peptide receptor radionuclide therapy. 

With many innovative approaches under investigation, it is hoped that a broader range of tumor-targeting PDCs will demonstrate therapeutic efficacy and improve outcomes for patients with difficult to treat or refractory cancers.

## Figures and Tables

**Figure 1 molecules-26-06042-f001:**
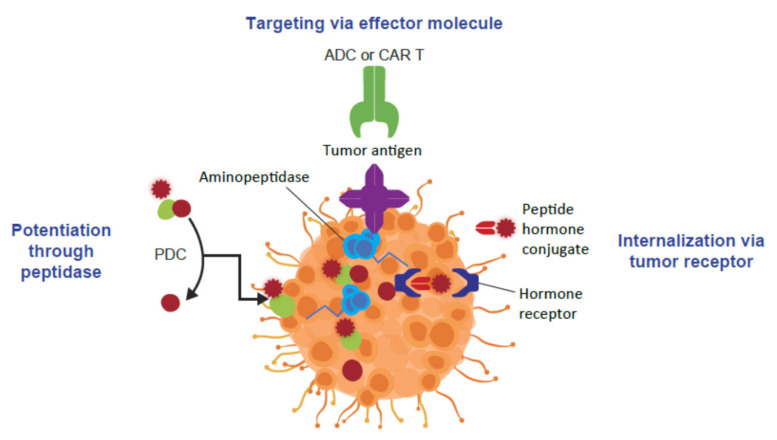
The different targeted approaches in cancer therapy. ADC: antibody-drug conjugate; CAR T: chimeric antigen receptor T cell; PDC: peptide-drug conjugate. Note that the aminopeptidase is intracellular.

**Figure 2 molecules-26-06042-f002:**
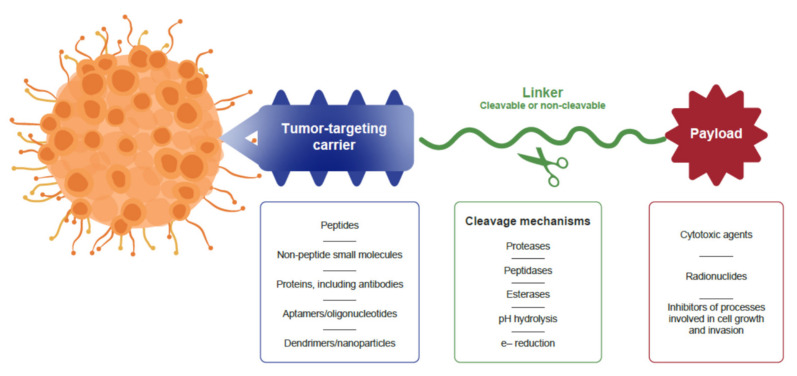
Structure of tumor-targeting drug conjugates. Adapted from Hoppenz et al. 2020 [1].

**Figure 3 molecules-26-06042-f003:**
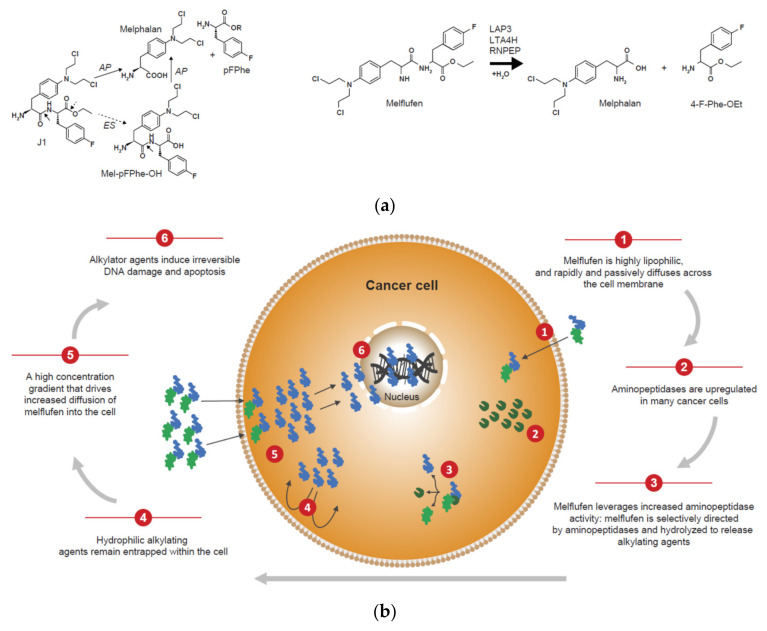
Structure and mechanism of action of melflufen, a first-in-class anti-cancer PDC: (**a**) structure of melflufen, a PDC with a highly lipophilic dipeptide carrier linked to the alkylating agent melphalan via an enzyme-cleavable linker; the cleavage of melflufen is mediated by both aminopeptidases and esterases; R: any group in which a carbon or hydrogen atom is attached to the rest of the molecule; (**b**) mechanism of action of melflufen, which is selectively recognized by the enzymes aminopeptidase (AP) and esterase (ES) overexpressed in cancer cells and hydrolyzed to release alkylating agents within the cell. Adapted from Wickström et al., 2017, Wickström et al., 2010, and Miettinen et al., 2021 [4,6,38].

**Figure 4 molecules-26-06042-f004:**
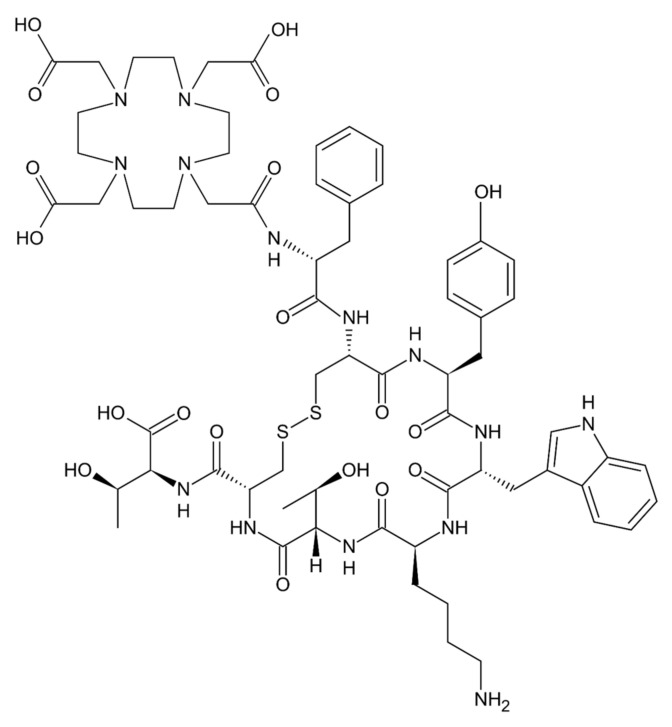
Chemical structure of dotatate. Image is freely available under the Creative Commons Attribution License (CC0).

**Table 1 molecules-26-06042-t001:** Drug conjugates currently approved by the FDA for clinical use as cancer treatments.

	Approved for: Tumor Type	Carrier Molecule	Linker	Payload	Molecular Weight
**Peptide-Drug** **Conjugates**					
Melflufen [18]	Multiple myeloma	Di-amino acid	Enzymatically cleaved after cellular uptake	Nitrogen mustard; 2-bischloroethyl amino (alkylating agent)	~0.5 kDa
^177^Lu-dotatate [19]	Gastroenteropancreatic neuroendocrine tumors	Somatostatin peptide analog	Covalently-bound chelator 1,4,7,10-tetraazacyclododecane-1,4,7,10-tetraacetic acid	Radionuclide ^177^Lutetium	1.6 kDa
**Antibody-Drug Conjugates**					
Ibritumomab tuixetan [20]	Non-Hodgkin lymphoma	Murine IgG_I_ kappa anti-CD20 (ibritumomab)	Tiuxetan [*N*-[2-bis(carboxymethyl)amino]-3-(p-isothiocyanatophenyl)-propyl]-[*N*-[2-bis(carboxymethyl)amino]-2-( methyl) -ethyl]glycine	Radionuclide ^111^Indium-Ill or ^90^Yttrium-90	148 kDa
^131^I-tositumomab [21]	Non-Hodgkin lymphoma	Murine IgG_2a_ lambda anti-CD20 (tositumomab)	Covalent bond	Radionuclide ^131^Iodine	150 kDa
Trastuzumab emtansine [22]	HER2-positive breast cancer	Humanized anti-HER2 IgG_1_ (trastuzumab)	Stable thioether (4-[N-maleimidomethyl] cyclohexane-1-carboxylate)	DM1 (microtubule inhibitor)	~148.5 kDa
Trastuzumab deruxtecan [23]	HER2-positive breast cancer and gastric or gastroesophageal junction adenocarcinoma	Humanized anti-HER2 IgG_1_ (trastuzumab)	Protease-cleavable maleimide tetrapeptide linker	DXd (topoisomerase inhibitor)	NA
Brentuximab vedotin [24]	Lymphoma	Chimeric anti-CD30 IgG_1_ antibody	Protease cleavable linker	Monomethyl auristatin E (microtubule disrupting agent)	153 kDa
Polatuzumab vedotin [25]	Diffuse large B-cell lymphoma	Humanized anti-CD79b IgG_1_	Protease-cleavable maleimidocaproyl-valine-citrulline-p-aminobenzyloxycarbonyl	Monomethyl auristatin E (microtubule disrupting agent)	150 kDa
Belantamab mafadotin [26]	Multiple myeloma	Afucosylated, humanized anti-B-cell maturation antigen IgG_1_	Protease-resistant maleimidocaproyl linker	Monomethyl auristatin F (microtubule disrupting agent)	152 kDa
Gemtuzumab ozogamicin [27]	Acute myeloid leukemia	Humanized anti-CD33 IgG_4_ kappa antibody	Covalent link	*N*-acetyl gamma calicheamicin (causes double-stranded DNA breaks)	NA
Inotuzumab ozogamicin [28]	Acute lymphoblastic leukemia	Humanized anti-CD22 IgG_4_ kappa antibody	Acid-cleavable linker	*N*-acetyl-gamma-calicheamicin (causes double-stranded DNA breaks)	160 kDa
Enfortumab vedotin [29]	Urothelial cancer	Human anti-nectin-4 IgG_1_ kappa antibody	Protease-cleavable maleimidocaproyl valine-citrulline linker	Monomethyl auristatin E (microtubule disrupting agent)	152 kDa
Sacituzumab govitecan [30]	Triple-negative breast cancer and urothelial cancer	Humanized anti-Trop-2 IgG_1_ kappa antibody	Hydrolysable linker (CL2A)	SN-38 (topoisomerase inhibitor)	160 kDa
Loncastuximab tesirine-lpyl [31]	B-cell lymphoma	Human anti-CD19 IgG_1_ kappa antibody	Protease-cleavable valine-alanine linker	SG3199 (pyrrolobenzodiazepine dimer cytotoxic alkylating agent)	151 kDa
Moxetumomab pasudotox-tdfk [32]	Hairy cell leukemia	Murine anti-CD22 Ig variable domain	Genetically fused	Truncated *Pseudomonas* exotoxin, PE38 (protein synthesis inhibitor)	63 kDa

FDA: Food and Drug Administration; HER2: human epidermal growth factor receptor-2; Ig: immunoglobulin; NA: not available.

**Table 2 molecules-26-06042-t002:** Rationally designed peptide-drug conjugates combining synthetic peptide analogs with cytotoxic chemotherapy.

PDC	Structure	Clinical Proof of Principle Studies
Zoptarelin doxorubicin [59]	Doxorubicin linked to a small peptide agonist to the LHRH receptor	Zoptarelin failed to improve efficacy or safety versus doxorubicin alone in Phase III, randomized study in patients with pre-treated advanced/metastatic recurrent endometrial cancer (NCT01767155)
GRN1005 [60,61]	Paclitaxel linked to angiopep-2, a peptide ligand that crosses the BBB through an LRP-1-mediated mechanism	GRN1005 delivers paclitaxel across the BBB and achieves therapeutic concentrations in tumor tissue Similar toxicity to paclitaxel Clinical activity in recurrent glioma Intracranial partial responses in 8% of patients with breast cancer and brain metastases (NCT01480583)
TH1902 [62]	Docetaxel linked to a peptide targeting sortilin (SORT1) receptors	Awarded FDA fast-track designation March 2021: first patient entered into Phase I study of TH1902 in patients with advanced solid tumors (NCT04706962)

BBB: blood–brain barrier; LHRH: luteinizing hormone-releasing hormone; LRP-1: low-density lipoprotein receptor-related protein-1.

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
