# Peer review of "Progress and Future Directions with Peptide-Drug Conjugates for Targeted Cancer Therapy"

_molecules, 2021, doi:10.3390/molecules26196042_

Round 1

Reviewer 1 Report

In this paper, Jakob Lindberg and co-workers review the peptide-drug conjugates (PDC) as a cytotoxic agent for cancer treatments. These are very meaningful. However, there are still some questions should be addressed before publication in the journal with intentional reputations.

My comments are as follows:

  1. As a review, part 4 “Peptide-Drug Conjugates as Targeted Cancer Treatments” has little introduction about the types of PDC, so it is suggested to systematic review the types of PDC drugs in part 4.
  2. It is recommended to supplement the contents of radioactive PDC drugs in part 4 “Peptide-Drug Conjugates as Targeted Cancer Treatments”.
  3. In part 2 “Structure of Anticancer Drug Conjugates”, the chemical formulas of Melflufen and 177Lu-dotatate were proposed to be added and explained.
  4. In keywords, “peptide-drug conjugates” and “PDC” seem to be the same mean.
  5. As a review, the references of PDC drugs is not enough, suggested cited more literature (the latest three years). https://doi.org/10.1016/j.ejmech.2020.113050;

https://doi.org/10.3390/biomedicines9080849

Author Response

Responses to reviewer 1 comments:

  1. As a review, part 4 “Peptide-Drug Conjugates as Targeted Cancer Treatments” has little introduction about the types of PDC, so it is suggested to systematic review the types of PDC drugs in part 4.

We have added further background on the types of PDC (please see yellow highlighted text on page 7).

  1. It is recommended to supplement the contents of radioactive PDC drugs in part 4 “Peptide-Drug Conjugates as Targeted Cancer Treatments”.

We have added a sentence about 131I-tositumomab (now included in Table 1) to supplement Section 4 (please see yellow highlighted text on pages 8 and 9). Based on recent reviews (Hoppenz 2020 and Cooper 2021, both cited in the manuscript), we do not believe that there are any additional approved radioconjugates to include. We have also defined the scope in the Introduction of this manuscript to provide some more detail and to emphasise that this review publication focuses on approved PDCs (please see yellow highlighted text on page 2). Also, it should be noted that as this manuscript is part of a special issue/edition, certain related topics that we have decided not to focus on will be covered more comprehensively in other publications/issues.

  1. In part 2 “Structure of Anticancer Drug Conjugates”, the chemical formulas of Melflufen and

177Lu-dotatate were proposed to be added and explained.

We have included chemical formulas and structures for both these products in Section 4 (please see yellow highlighted text on pages 7 and 8, and new figure 4 [for 177Lu-dotatate] on page 9 – structure for melflufen is already present in figure 3a).

  1. In keywords, “peptide-drug conjugates” and “PDC” seem to be the same mean.

We have removed ‘PDC’ from keywords list (please see yellow highlighted text on page 1).

  1. As a review, the references of PDC drugs is not enough, suggested cited more literature (the latest three years).https://doi.org/10.1016/j.ejmech.2020.113050; https://doi.org/10.3390/biomedicines9080849.

We have adjusted the title of this review article accordingly. We have also defined the scope of the manuscript in the introduction with more detail to emphasise that this review publication focuses on approved PDCs (please see yellow highlighted text on page 2). We have also included a few selected additional examples of PDCs under development. From reading some recent PDC reviews, including the recommended papers from reviewer, we do not feel that we have omitted any key PDCs in this publication. However, we have added some further text on the PDCs TH1902 and TH1904 from Theratechnologies, which are currently in clinical and pre-clinical development, respectively (please see yellow highlighted text on page 10; however, please note that TH1902 was already covered in Table 2, as well as BTCs in early clinical development, e.g. BT5528). We have also added some text on self-assembling PDCs (please see yellow highlighted text on page 12).

Reviewer 2 Report

The manuscript describes recent progress in the development of peptide drug conjugate. The subject is very interesting and has attracted increasing attention over the past decades. The authors attempted to provide a rational comparison between antibody-drug conjugate and peptide drug conjugate, and in the end, summarized their clinical applications and offered perspectives on their future growth. Overall, the manuscript is very organized, and the chosen topic is significant. However, the authors seemed to focus on a very narrow area, and lots of exciting progress made in the design, synthesis, and assessment of peptide drug conjugates have been left out. The manuscript will be more impactful by including this work. Please check the review article in ADDR ( Wang et al.  Peptide-drug conjugates as effective prodrug strategies for targeted delivery, Advanced Drug Delivery Reviews, 2017, 110-111, 112-126.) and the references therein. The recent work on self-assembling camptothecin and paclitaxel drug peptide conjugates has been gaining lots of attention, and the authors are encouraged to consider these manuscripts. The targeting strategy is passive, by taking advantage of the EPR effect of tumor pathology.  

Author Response

Responses to reviewer 2 comments:

  1. The manuscript describes recent progress in the development of peptide drug conjugate. The subject is very interesting and has attracted increasing attention over the past decades. The authors attempted to provide a rational comparison between antibody-drug conjugate and peptide drug conjugate, and in the end, summarized their clinical applications and offered perspectives on their future growth. Overall, the manuscript is very organized, and the chosen topic is significant. However, the authors seemed to focus on a very narrow area, and lots of exciting progress made in the design, synthesis, and assessment of peptide drug conjugates have been left out. The manuscript will be more impactful by including this work. Please check the review article in ADDR ( Wang et al. Peptide-drug conjugates as effective prodrug strategies for targeted delivery, Advanced Drug Delivery Reviews, 2017, 110-111, 112-126.) and the references therein. The recent work on self-assembling camptothecin and paclitaxel drug peptide conjugates has been gaining lots of attention, and the authors are encouraged to consider these manuscripts. The targeting strategy is passive, by taking advantage of the EPR effect of tumor pathology.

We have added some text on self-assembling PDCs (please see yellow highlighted text on page 12). Also, it should be noted that as this manuscript is part of a special issue/edition, certain related topics that we have decided not to focus on will be covered more comprehensively in other publications/issues.

Round 2

Reviewer 1 Report

The points raised were addressed and the manuscript can be suitable for publication in the present form.